# Genetic substructure and complex demographic history of South African Bantu speakers

Dhriti Sengupta[1,15], Ananyo Choudhury [1,15], Cesar Fortes-Lima [2], Shaun Aron[1], Gavin Whitelaw [3,4], Koen Bostoen [5], Hilde Gunnink[5], Natalia Chousou-Polydouri[6], Peter Delius[7], Stephen Tollman [8], F. Xavier Gómez-Olivé[8], Shane Norris [9], Felistas Mashinya [10], Marianne Alberts[10], AWI-Gen Study*, H3Africa Consortium*, Scott Hazelhurst [1,11], Carina M. Schlebusch [2,13,14,16] & Michèle Ramsay [1,12,16 ✉]

South Eastern Bantu-speaking (SEB) groups constitute more than 80% of the population in South Africa. Despite clear linguistic and geographic diversity, the genetic differences between these groups have not been systematically investigated. Based on genome-wide data of over 5000 individuals, representing eight major SEB groups, we provide strong evidence for fine-scale population structure that broadly aligns with geographic distribution and is also congruent with linguistic phylogeny (separation of Nguni, Sotho-Tswana and Tsonga speakers). Although differential Khoe-San admixture plays a key role, the structure persists after Khoe-San ancestry-masking. The timing of admixture, levels of sex-biased gene flow and population size dynamics also highlight differences in the demographic histories of individual groups. The comparisons with five Iron Age farmer genomes further support genetic continuity over ~400 years in certain regions of the country. Simulated trait genome-wide association studies further show that the observed population structure could have major implications for biomedical genomics research in South Africa.

[1] Sydney Brenner Institute for Molecular Bioscience, Faculty of Health Sciences, University of the Witwatersrand, Johannesburg, South Africa. [2] Human Evolution, Department of Organismal Biology, Evolutionary Biology Centre, Uppsala University, Uppsala, Sweden. [3] KwaZulu-Natal Museum, Pietermaritzburg, South Africa. [4] School of Geography, Archaeology & Environmental Studies, University of the Witwatersrand, Johannesburg, South Africa. [5] UGent Centre for Bantu Studies, Department of Languages and Cultures, Ghent University, Ghent, Belgium. [6] Department of Comparative Linguistic Science and Center for the Interdisciplinary Study of Language Evolution, University of Zürich, Zürich, Switzerland. [7] Department of History, University of the Witwatersrand, Johannesburg, South Africa. [8] MRC/Wits Rural Public Health and Health Transitions Research Unit (Agincourt), School of Public Health, Faculty of Health Sciences, University of the Witwatersrand, Johannesburg, South Africa. [9] MRC/Wits Developmental Pathways for Health Research Unit, Faculty of Health Sciences, University of the Witwatersrand, Johannesburg, South Africa. [10] Department of Pathology and Medical Sciences; School of Health Care Sciences, Faculty of Health Sciences, University of Limpopo, Polokwane, South Africa. [11] School of Electrical and Information Engineering, University of the Witwatersrand, Johannesburg, South Africa. [12] Division of Human Genetics, National Health Laboratory Service and School of Pathology, Faculty of Health Sciences, University of the Witwatersrand, Johannesburg, South Africa. [13] SciLifeLab, Uppsala, Sweden. [14] Palaeo-Research Institute, University of Johannesburg, Johannesburg, South Africa. [15] These authors contributed equally: Dhriti Sengupta, Ananyo Choudhury. [16] These authors jointly supervised this work: Carina M. Schlebusch, Michèle Ramsay. *Lists of authors and their affiliations appear at the end of the paper. ✉email: Michele.Ramsay@wits.ac.za

The sarchaeological record and rock art evidence trace the presence of San-like hunter-gatherer culture in Southern Africa to at least 20–40 thousand years ago (kya)[1–3]. Three sets of migration events have dramatically reshaped the genetic landscape of this geographic region in the last two millennia. The first of these was a relatively small scale migration of East African pastoralists, who introduced pastoralism to Southern Africa ~2 kya[4–7]. This population was subsequently assimilated by local Southern African San hunter-gatherer groups, forming a new population that was ancestral to the Khoekhoe herder populations[8–12]. Today, Southern African Khoe and San populations collectively refer to hunter-gatherer (San) and herder (Khoekhoe) communities. While Khoe-San groups are distributed over a large geographic area today (spanning the Northern Cape Province of South Africa, large parts of Namibia, Botswana, and Southern Angola), these groups are scattered, small, and marginalised[13,14].

The introduction of pastoralism in the region was closely followed by the arrival of the second set of migrants i.e., the Bantu-speaking (BS) agro-pastoralists. The archaeological record suggests that ancestors of the current-day BS populations undertook different waves of migration instead of a single large-scale movement[15–17]. The earliest communities spread along the East coast to reach the KwaZulu-Natal South coast by the mid-fifth century AD while the final major episode of settlement is estimated to be around AD 1350[18,19]. These archaeologically distinct groups gradually spread across present-day South Africa, interacting to various degrees with the Khoe-San groups, eventually giving rise to South Africa's diverse BS communities. The third major movement into Southern Africa was during the colonial era in the last four centuries when European colonists settled the area. During this period slave trade introduced additional intercontinental gene flow giving rise to complex genomic admixture patterns in current-day Southern African populations[20–23].

South Africa has 11 official languages of which nine are Bantu languages belonging to this family's South-Eastern branch. Within these nine languages two large subclusters are traditionally distinguished: Nguni (including Zulu, Xhosa, Swazi, and Ndebele) and Sotho-Tswana (including Sotho, Tswana, and Pedi). Venda and Tsonga tend to be seen as independent linguistic entities[24–27]. A new lexicon-based linguistic phylogeny included in this study (Supplementary Note 1) broadly confirms the traditionally recognized clusters, but also adds possible insights into how these languages might relate to each other as well as to 60 other Bantu languages. While the genetic diversity of Khoe-San and mixed ancestry groups has been widely investigated[28], the genetic diversity of the SEB-speaking (referred henceforth as SEB) groups has not been systematically investigated. One of the very early studies based on the Y-chromosome and a few autosomal markers, which included almost all the main SEB groups and covered most of the provinces from South Africa, indicated the possibility of genetic structure within the SEB populations[29]. However, many subsequent studies using genome-wide datasets did not investigate genetic differentiation or population structure within SEB groups, which consequently led to its consideration as a group without clear internal substructure[21,30]. Moreover, studies including multiple SEB groups were often limited in terms of sample size or SEB group diversity[22,31,32].

Here we describe a systematic study of genetic diversity of South African SEB groups based on an analysis of 5056 individuals (AWI-Gen study) genotyped on the Illumina H3A-genotyping array (~2.3 M SNPs). Although the eight SEB groups have very specific geographic distributions of linguistic majority areas (LMAs) within the country, for our study they were sampled at three sites; Soweto (SWT) in Gauteng, Agincourt (AGT) in Mpumalanga, and Dikgale (DKG) in Limpopo province (Table 1 and Fig. 1a).

This study reports a fine-scale population structure among SEB groups that parallels both linguistic affinities and geographic distribution in the country. Simulated association studies demonstrate that the fine-scale structure has a potential to influence genetic association in cohorts that include multiple SEB groups and/or SEB groups from multiple study sites within South Africa.

## Results

**Fine-scale population structure within SEB**. The principal component analysis (PCA) of 4319 unrelated SEB participants (AWI-S2 dataset) reflects the linguistic phylogeny with partial separation of Tsonga, Sotho-Tswana (Sotho, Pedi, and Tswana) and Nguni (Zulu and Xhosa) speakers (Fig. 1b, Supplementary Fig. 1 and Supplementary Note 1). The distribution of the SEB groups on the PC plot also largely mirrors the LMAs of these groups on the South African map (Fig. 1a, b) suggesting a correlation between genetic variation and geography.

The movement of populations from their LMAs to other regions during the last century is known to have enhanced the genetic exchange between different SEB groups, especially in urban areas such as Soweto[33]. These recent admixtures could result in incomplete boundaries observed between the SEB groups in the PC plot. In order to minimize the effect of such recent admixture on population structure detection, we analysed 2702 SEB participants (Fig. 1c), who self-reported to share the same ethno-linguistic identity for at least five of the six parents and grandparents. We refer to these individuals as ethno-linguistically concordant (EC) participants hereafter (AWI-S3 dataset) (Table 1). The EC-based filtering step (Table 1) enhanced the resolution of these groups on the PC analysis and also reduced the number of participants clustering with a different SEB group (Fig. 1c). PCA-UMAP analysis[34], based on a composite of the first 10 PC coordinates estimated using EC participants, further illustrates the separation between the SEB groups (Fig. 1d). To avoid the likely influence of sample size-bias, we randomly downsized each group (AWI-S4 dataset). Likewise, both the PC and PCA-UMAP plots for this downsized data largely retained the fine-scale structure within SEB groups (Supplementary Fig. 2a, b). In addition, we performed haplotype-based analysis on the basis of the AWI-S4 dataset using ChromoPainter/fineSTRUCTURE[35] (see Supplementary Note 2). Haplotype-based PCA and the pairwise-coincidence matrix among EC individuals provides further support for the fine-scale population structure among SEB groups (Supplementary Fig. 2c, d). These results highlight the importance of ethno-linguistically informed sampling for inferring the fine-scale population structure and also provides a possible rationale for why some previous studies, especially based on individuals from urban centres, could have underestimated population structure in SEB groups.

We compared our SEB populations to previously studied populations from Southern Africa[21,31,32,36] by performing PC analysis with Merged dataset 2 (Supplementary Table 1). The PC plot shows Zulu, Xhosa, and Sotho individuals from these studies to group with corresponding SEB groups from the AWI-Gen study (Supplementary Fig. 1a, b). Similarly, some of the individuals from Mozambique[36] clustered close to Tsonga and Venda from our dataset, indicating the population structure to be largely robust. Phylogenetic trees based on genetic distances ($F_{ST}$) (Fig. 1e and Supplementary Fig. 3a) and linguistic phylogeny (Fig. 1f, Supplementary Fig. 3b, and Supplementary Note 1) of the SEB groups shows overall alignment in topology. Similarly, the genetic ($F_{ST}$) and geographical distances between the SEB groups also show a moderate correlation (Mantel test r value: 0.56, P-value = 0.002). Procrustes transformation analysis further

**Table 1 Distribution of the South-Eastern Bantu-speaking (SEB) group by centre and ethnicity.**

| SEB group | Agincourt centre | | | Dikgale centre | | | Soweto centre | | | Total | | |
|---|---|---|---|---|---|---|---|---|---|---|---|---|
| | All | UR | EC | All | UR | EC | All | UR | EC | All | UR | EC |
| Pedi | 36 | 33 | 0 | 1106 | 924 | 812 | 109 | 108 | 39 | 1251 | 1065 | 851 |
| Sotho | 97 | 79 | 0 | 9 | 9 | 5 | 285 | 278 | 41 | 391 | 366 | 46 |
| Swazi | 88 | 70 | 19 | 2 | 1 | 0 | 56 | 55 | 11 | 146 | 126 | 30 |
| Tsonga | 1941 | 1487 | 1369 | 52 | 47 | 22 | 117 | 110 | 47 | 2110 | 1644 | 1438 |
| Tswana | 1 | 1 | 1 | 14 | 13 | 5 | 234 | 228 | 67 | 249 | 242 | 73 |
| Venda | 5 | 5 | 2 | 23 | 21 | 6 | 47 | 47 | 16 | 75 | 73 | 24 |
| Xhosa | 3 | 3 | 2 | 6 | 6 | 4 | 169 | 168 | 57 | 178 | 177 | 63 |
| Zulu | 58 | 46 | 12 | 10 | 8 | 5 | 588 | 572 | 160 | 656 | 626 | 177 |
| Total | 2229 | 1724 | 1405 | 1222 | 1029 | 859 | 1605 | 1566 | 438 | 5056 | 4319 | 2702 |

The three columns for each centre shows: the total number of samples (All), the number of unrelated samples (PIHAT < 0.18) (UR) and the ethno-linguistically concordant (EC) samples (self-reported ethno-linguistic identity of a participant is same as the ethno-linguistic identity of at least five of the six parents and grandparents). The column "All" corresponds to the AWI-S1, "UR" corresponds to AWI-S2 and "EC" corresponds to AWI-S3 dataset (Supplementary Table 1).

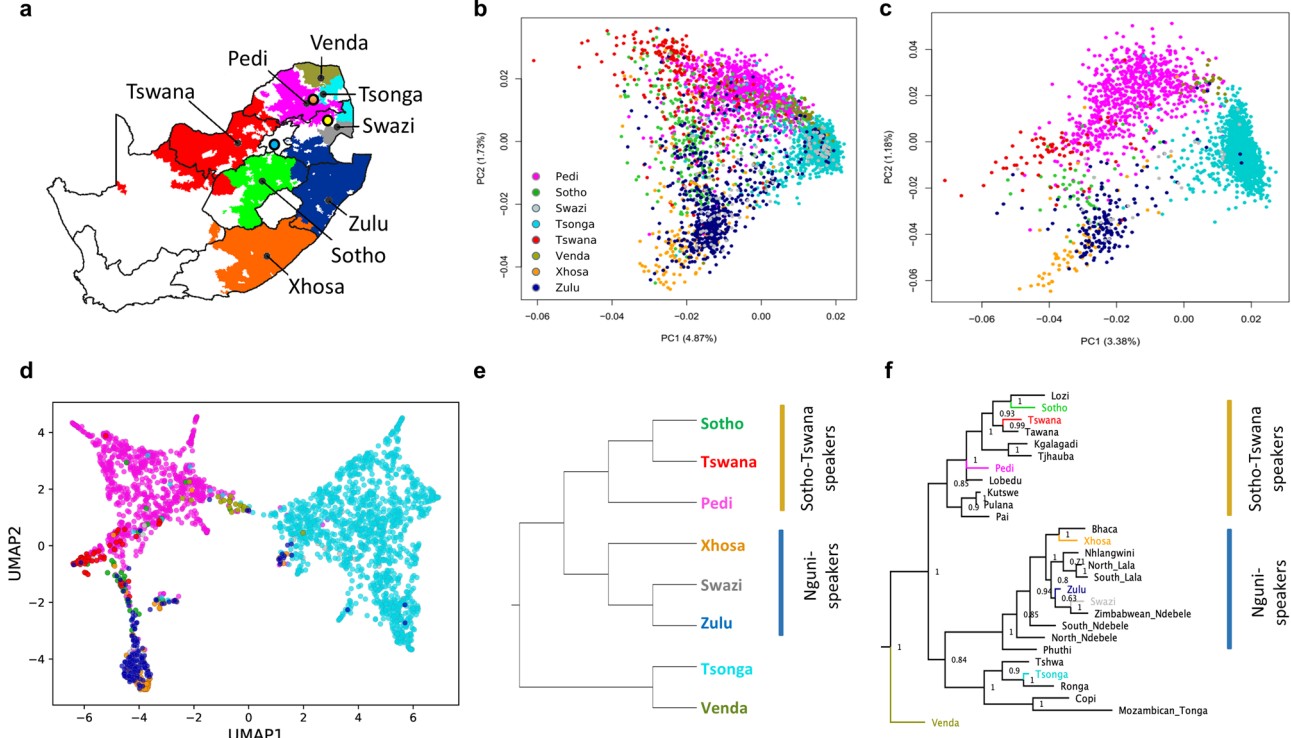

**Fig. 1 Population structure and genetic affinities of South-Eastern Bantu-speaking (SEB) groups from South Africa correspond to both linguistic phylogeny and geographic distribution. a** Map showing the language majority areas (LMAs) of each SEB group. The centroid of each of the regions is indicated using a black dot. The three sampling sites are shown in coloured circles; Soweto in blue, Dikgale in orange and Agincourt in yellow. The original map was obtained from: https://en.wikipedia.org/wiki/Languages_of_South_Africa#/media/File:South_Africa_2011_dominant_language_map.svg. The user acknowledges Stats SA as the source of the basic data wherever they process, apply, utilise, publish or distribute the data, and also that they specify that the relevant application and analysis (where applicable) result from their own processing of the data. The language centroid points were calculated for this study (see methods for details). **b** Principal Component (PC) plot for the unrelated SEB samples (Pedi $N = 1065$, Sotho $N = 366$, Swazi $N = 126$, Tsonga $N = 1644$, Tswana $N = 242$, Venda $N = 73$, Xhosa $N = 177$ and Zulu $N = 626$) shows an overall correspondence between the distribution of SEB groups on the geographic map and the PCA. The colours showing the LMA for each SEB group on the geographic map corresponds to the colours used for the SEB group in the PCA. **c** PC plot based on ethno-linguistically concordant samples (self-reported ancestry of the participant is the same as at least 5 of the parents and grandparents) (Pedi $N = 851$, Sotho $N = 46$, Swazi $N = 30$, Tsonga $N = 1438$, Tswana $N = 73$, Venda $N = 24$, Xhosa $N = 63$ and Zulu $N = 177$) shows much clearer separation between the three major linguistic divisions (Sotho-Tswana, Nguni, and Tsonga speakers). **d** A composite representation of the first 10 PCs (generated using PCA-UMAP) also shows separation of the SEB groups corresponding to the three major linguistic divisions. **e** UPGMA tree based on pairwise $F_{ST}$ distance between SEB groups. Sample sizes are same as of panels **c**. **f**, Linguistic phylogeny based on lexical data (majority-rule consensus tree) with posterior probability values. The SEB groups from the current study are indicated using the same colours as used in the PCA plots. The topology of the trees in **e** and **f** shows an overall alignment.

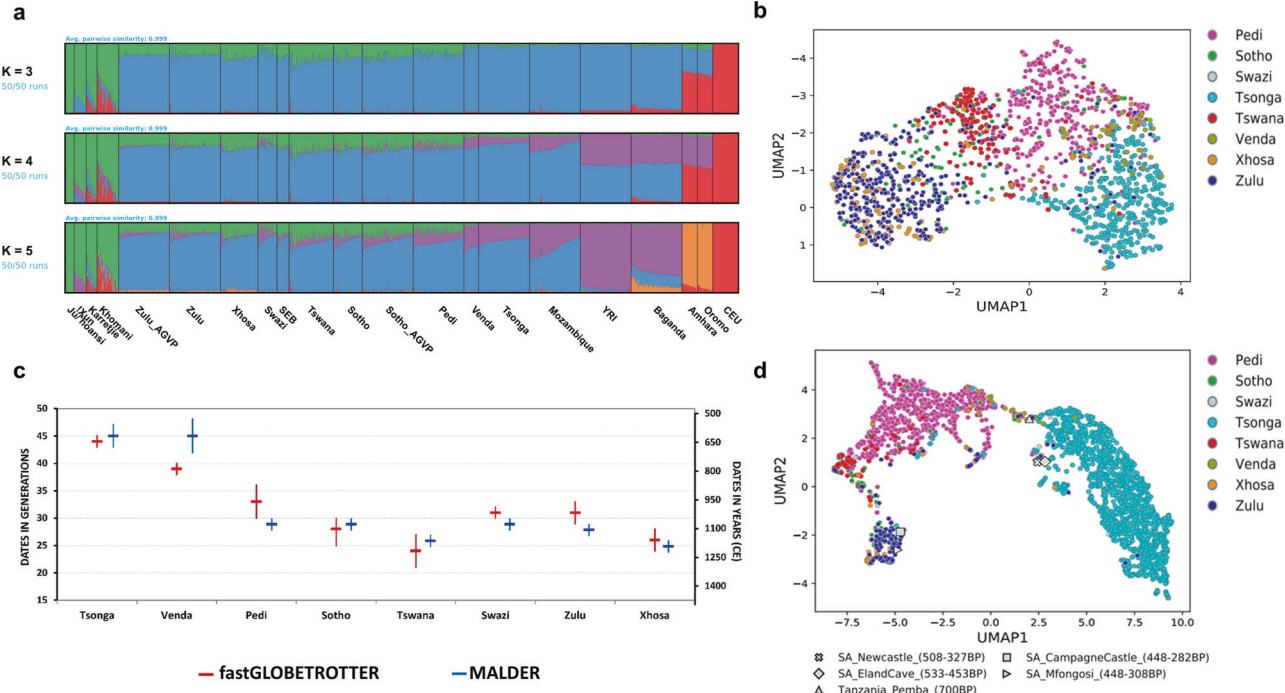

**Fig. 2 Gene flow into and genetic continuity of South-Eastern Bantu-speaking (SEB) groups. a** ADMIXTURE plots (from $K = 3$ to $K = 5$) based on the merged dataset with downsized ethno-linguistically concordant individuals (Pedi $N = 80$, Sotho $N = 45$, Swazi $N = 30$, Tsonga $N = 80$, Tswana $N = 70$, Venda $N = 23$, Xhosa $N = 59$, Zulu $N = 80$, Sotho_AGVP $N = 80$, Zulu_AGVP $N = 80$, Mozambique $N = 80$, SEB $N = 19$, Amhara $N = 24$, Oromo $N = 24$, Baganda $N = 80$, YRI $N = 80$, CEU $N = 80$, Jul'hoansi $N = 14$, Karretjie $N = 17$, !Xun $N = 19$ and Khomani $N = 34$). At $K = 3$, the plot shows differences in the level of Khoe-San gene flow (shown in green) into different SEB groups, with Tswana and Xhosa showing the highest Khoe-San ancestry proportion and Tsonga and Venda the lowest. Baganda (from Uganda); Amhara, Oromo and Somali (from Ethiopia); Sotho_AGVP and Zulu_AGVP (from South Africa) are from (ref. [32]) datasets. The Yoruba (YRI) and Central European (CEU) are from the 1000 Genomes Project dataset[61]. **b** Composite representation of the first 10 PCs (generated using ancestry-specific PCA-UMAP) showing population structure in SEB groups persists even after Khoe-San ancestry masking. Sample sizes are same as of Fig. 1c. **c** Dates for Khoe-San admixture in SEB populations estimated using fastGLOBETROTTER (red dates) and MALDER (blue dates). Figure also showing 95% CI bars (vertical lines) from each method. First y-axis shows admixture dates in generations ago, while the second y-axis shows the actual estimated dates. Confidence intervals (95% CI) of estimates of dates were based on 50 bootstrap replicates for each population in each admixture dating analysis. CE refers to the Common Era. **d** Composite representation of the first 10 PCs comparing Iron-Age genomes to our SEB groups indicate genetic continuity for the last few centuries in certain regions of South Africa. Sample sizes are same as of Fig. 1c.

highlights the correlation between PC and geography ($r^2 = 0.72$; $P$-value $= 0.0009$) (Supplementary Fig. 3c). However, the overall low magnitude of $F_{ST}$ values (Supplementary Fig. 3d) suggests that the fine-scale structure, although robust, corresponds to relatively small genetic distances.

**Differential Khoe-San gene flow into various SEB groups.** As Khoe-San gene flow has been reported to be a major factor in differentiating SEB groups[22,31,32], we estimated the level of Khoe-San ancestry proportions in each SEB group (based on the Merged dataset 2-EC downsized) using an unsupervised clustering approach[37]. ADMIXTURE analysis at $K = 3$ highlights the separation of BS, Khoe-San and Eurasian ancestry (blue, green, and red component, respectively) (Fig. 2a). The various SEB groups showed differential levels of Khoe-San gene flow varying from $1.5 \pm 2\%$ in Tsonga to $20 \pm 6\%$ in Tswana (Table 2). The lowest cross-validation value was observed at $K = 5$, which separates the Afro-Asiatic and the Central-West African ancestries (Fig. 2a and Supplementary Fig. 4a). To investigate the impact of differential sample sizes from Eastern, Western, and Southern Africa in these estimates, we also performed the ADMIXTURE analysis using a dataset with a more uniform representation of samples from the three regions (Supplementary Fig. 4b).

ADMIXTURE analysis on the full set of unrelated samples (Merged dataset 1; details in Supplementary Table 1) detected

**Table 2 Ancestry proportions for various South-Eastern Bantu-speaking (SEB) groups estimated using unsupervised ADMIXTURE analysis (at $K = 3$).**

| SEB group | Sample size | Bantu-related ancestry (%) | | Khoe-San-related ancestry (%) | | Eurasian-related ancestry (%) | |
|---|---|---|---|---|---|---|---|
| | | Mean | ±SD | Mean | ±SD | Mean | ±SD |
| Pedi | 1065 | 88.28 | 5.11 | 10.61 | 4.71 | 1.12 | 2.83 |
| Sotho | 366 | 84.17 | 8.36 | 14.65 | 7.40 | 1.18 | 3.60 |
| Tswana | 242 | 78.19 | 7.44 | 20.49 | 6.02 | 1.32 | 4.59 |
| Swazi | 126 | 90.43 | 8.02 | 8.69 | 7.49 | 0.87 | 2.59 |
| Xhosa | 177 | 80.24 | 5.90 | 17.62 | 4.86 | 2.14 | 3.17 |
| Zulu | 626 | 84.64 | 6.32 | 13.58 | 4.72 | 1.78 | 4.16 |
| Tsonga | 1644 | 97.80 | 2.86 | 1.56 | 2.43 | 0.65 | 1.21 |
| Venda | 73 | 91.31 | 6.99 | 6.45 | 5.15 | 2.24 | 4.16 |

considerable within-population variation in ancestry proportions for some of the SEB groups (Supplementary Fig. 4c). When partitioned by the study site, four of the SEB groups (Zulu, Sotho, Pedi, and Swazi) show significantly higher Khoe-San ancestry in participants originating from SWT in comparison to participants from AGT (Supplementary Fig. 4d-g, and Supplementary Table 2). These differences for populations such as Swazi and Zulu were also distinguishable in a PC plot that includes the site of collection information along with group labels (Supplementary

Fig. 5). These observations emphasize the importance of careful consideration of sampling locations in addition to ethno-linguistic concordance, for a comprehensive estimation of the fine-scale population structure. Notably, the estimates show about 170 (4% on average) of the SEB participants harbour more than 5% Eurasian-like ancestry (Table 2). As there has been no systematic study to estimate the level of Eurasian ancestry in the more Northern provinces of the country, we were unable to estimate whether the observed level of Eurasian ancestry is common in SEB groups from these geographic areas or a cohort-specific feature. Our results could therefore provide a baseline for future studies on Eurasian admixture in SEB groups.

To further investigate whether differential Khoe-San gene flow was the only factor leading to the observed population structure, we filtered out non-Bantu-related haplotypes in each SEB individual (see Methods). Ancestry-specific PCA after masking haploid genomes shows that the core differences within SEB groups, although reduced, persist even after accounting for differential Khoe-San gene flow (Fig. 2b and Supplementary Fig. 6). The observed structure between SEB groups could therefore be attributed to additional historical and demographic factors, such as multiple expansion movements into Southern Africa, different points of origin and isolation due to geography.

**Dating admixture events in SEB groups.** To reconstruct the timeline of migration of each SEB group, we dated the admixture between the best BS and Khoe-San source populations for each group using fastGLOBETROTTER[38] (Fig. 2c and Supplementary Table 3). As the range of Khoe-San populations is estimated to have been much wider in the past compared to their present distribution, some of these admixture events might have occurred beyond the boundaries of the country. Moreover, it is also possible that in some cases gene flow from the Khoe-San might not have immediately followed the arrival of the BS populations. Nevertheless, it is reasonable to expect that major differences in admixture dating could be broadly indicative of the differences in dates of arrival and settlement of the ancestral SEB group in different regions of the country.

Consistent with many previous studies[22,32,39,40], the inferred dating pattern indicates that the contact between the ancestors of all the SEB groups and Khoe-San populations included in our study, occurred within the last 45 generations (~1300 years). Moreover, for all SEB groups, a single admixture event model was detected to be the best-guess conclusion by fastGLOBETROTTER (Supplementary Note 2). Tsonga and Venda show the oldest admixture dates (around 45 generations ago) while the admixture dates for the other SEB groups range between 24 and 33 generations ago. The presence of SEB groups on the South African landscape is assumed to date back to the fourth century AD, from which time there is considerable archaeological evidence for interaction with Khoe-San that probably included admixture[41]. The admixture dates for Tsonga and Venda, therefore, suggest that these SEB groups of southern Mozambique and North-Eastern South Africa could be descendants from one of the earlier episodes of settlement in this region.

The admixture dates correlate broadly with geography, with more Northern populations showing relatively older dates compared to Southern populations, for example, Zulu compared to Xhosa (Fig. 2c and Supplementary Table 3). Even among the groups from the inland plateau region (referred to as the highveld), we observed more recent dates for more Southern/Western populations (the Sotho and Tswana), compared to the more Northern Pedi. However, we also observed exceptions to these trends, such as a large difference between the Khoe-San admixture dates in geographically neighbouring Pedi and Tsonga.

Multiple westward movements of Tsonga speakers from Mozambique in the last few centuries have been reported[42] suggesting that the Tsonga and Pedi might have been separated by much greater geographic distances in the past, likely explaining the stark differences in admixture dating.

To test the robustness of the observed dating patterns, we also dated these admixture events using MALDER[43] (Fig. 2c) and MOSAIC[44] (Supplementary Table 3 and Supplementary Note 2). Although there are some differences in the predicted time-scales of admixture events obtained using these dating methods (MOSAIC for most groups generated younger dates), all the admixture dating methods demonstrate the same pattern (Supplementary Table 3, Supplementary Fig. 7, and Supplementary Note 2). The suggested dates for BS gene flow into Khoe-San populations, especially in the southern Kalahari region appears to be much younger (10–15 generations ago)[45,46] compared to the estimated dates for Khoe-San gene flow into the SEB. These large-scale differences in dates indicate the possibility of independent migration and admixture dynamics of BS in eastern and western Southern Africa. The estimated dates of Eurasian admixture in SEB groups (4–5 generations ago, Supplementary Table 4) is consistent with the rather recent settlement of European ancestry populations in the geographic region corresponding to the three sampling sites[47].

**Relationship between ancient genomes and modern SEB groups.** The availability of Iron-Age genomes from Southern Africa provided us with the unique opportunity to compare the affinities of present-day SEB groups to populations living in these areas centuries ago[10,11]. The PCA and PCA-UMAP projecting five Iron-Age Bantu-related genomes (300–700 years old) onto the genetic variation of present-day SEB individuals (Fig. 2d and Supplementary Fig. 8) show these genomes to be on a temporal cline with the older genomes (Pemba, Eland Cave and Newcastle; ranging ~700–450 BP) appearing closer to the Tsonga and Venda, while more recent genomes (Champagne Castle and Mfongosi; ranging from ~448–300 BP) occurring closer to the Nguni-speakers. This cline of the Iron-Age genomes also aligns with geographic distribution from North to South, as well as increasing levels of Khoe-San ancestry in them[10]. More ancient genomes from Southern Africa would be required to test whether the trends observed in these Iron-Age genomes are indicative of phases in the movement of groups further south with time, a process marked by concomitant increase of Khoe-San ancestry in the migrants. Interestingly, the wider geographic region of Northern KwaZulu-Natal around Champagne Castle in Central-East South Africa, where the youngest of these Iron-Age genomes was collected, is still dominated by Nguni-speakers (Fig. 1a), providing support for at least four centuries of genetic continuity in certain regions of South Africa.

**Sex-specific admixture patterns.** In accordance with several previous reports[31,48,49], the comparison of mitochondrial DNA and Y-chromosome haplogroup distributions (Fig. 3a, Supplementary Table 5, and Supplementary Table 6) shows evidence for relatively higher maternal gene flow from Khoe-San into all the SEB groups (Supplementary Note 3). Comparison of the autosomal and X-chromosome contributions also supports Khoe-San biased maternal gene flow (Fig. 3b). However, the level of this bias varies widely between groups (Fig. 3a, b, and Supplementary Table 7). The lack of any correlation between the extent of this bias and level of admixture/admixture dates suggests that the nature of interaction between Khoe-San and BS could have been determined by other demographic factors (Supplementary Note 3).

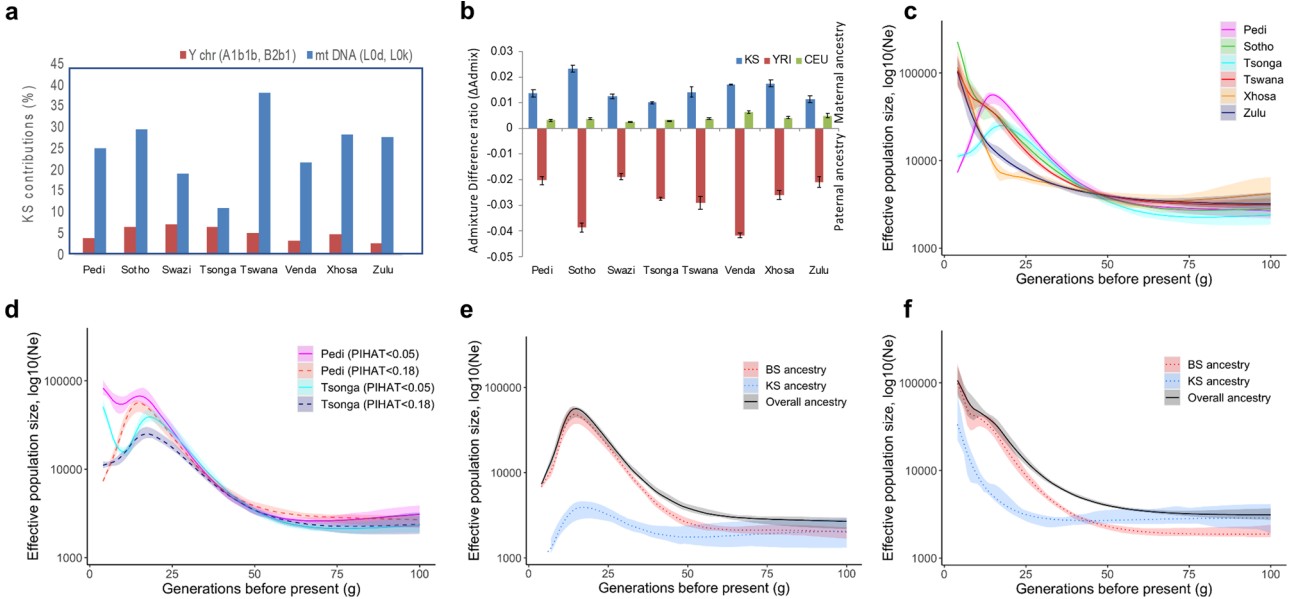

**Fig. 3 Insights into the demographic history of South-Eastern Bantu-speaking (SEB) groups. a** Distribution of Khoe-San (KS) associated mitochondrial and Y-chromosome haplogroups in the SEB groups shows higher maternal contribution from Khoe-San. **b** The analysis of admixture difference ratio (based on X chromosomal and autosomal contributions) confirms this trend and shows the level of bias to vary strongly between the SEB groups. The bars show admixture differences for the three contributing ancestries. Blue shows Khoe-San, red shows Bantu-speaker (represented by KGP Yoruba (YRI)) and green shows Eurasian (represented by KGP Central European (CEU)) ancestries for each SEB group. Positive bar values denote a maternal bias whereas negative values denote paternal bias in contributions from an ancestry. The error bars are based on 50 bootstrapping iterations with 20 samples each (source data provided in Source Data file). **c** Effective population size (*Ne*) fluctuations (estimated using IBDNe) shows SEB groups to differentiate mainly in the last 40 generations. **d** *Ne* profile differences in Pedi and Tsonga before and after removal of individuals with 0.05<PIHAT < 0.18. **e**, **f** Ancestry-specific IBDNe based evaluation of the relative contribution of Khoe-San and BS to the *Ne* profiles in **e** Pedi, and **f** Tswana. For **e** and **f**, the black line shows the overall ("true") *Ne* while the red and blue lines show the *Ne* for BS and Khoe-San ancestral components, respectively. The plots show the level of Khoe-San ancestry to correlate with the extent of influence on overall *Ne*. For **c–f**, the lines represent maximum likelihood inference, with shaded regions demarcating 95% confidence intervals based on 80 bootstrapping runs.

**Variation of effective population size through time**. We investigated changes in the effective population size (*Ne*) of each SEB group over the last 100 generations by analysing the sharing patterns of identity-by-descent (IBD) segments using IBDNe[50]. As depicted in Fig. 3c, the *Ne* for all the SEB groups was very similar for the 100th to the 40th generations before present. It needs to be noted that most of the present-day SEB groups did not exist, as such, more than 50 generations ago and the older estimates here correspond to possible ancestral populations of these groups. The period of around 40 generations ago also corresponds to the estimated time scale for the oldest Khoe-San admixture dates (Fig. 2c). From the 40th generation onwards, the Nguni-speakers and Sotho-Tswana speakers start showing distinct and characteristic *Ne* profiles, which possibly reflect migration events that separated these populations in terms of geography. Similarly, the dates for the initiation of population size increase of the Zulu around 25 generations ago, broadly corresponds to the time (around AD 1300) when Nguni-speakers first began to move North-west into the interior, becoming the first BS in South Africa to occupy grasslands[6,51]. The comparison of Sotho and Zulu *Ne* profiles between our study and samples from a previous study[32] shows a high concordance, demonstrating an overall robustness in these estimates (Supplementary Fig. 9).

A high level of cryptic relatedness in a population could strongly impact estimates based on IBD-sharing. Despite adopting a sampling strategy aimed at minimizing the recruitment of genetically related participants, we observed very high levels of cryptic relatedness in Tsonga and Pedi (Fig. 3d, Supplementary Fig. 10, and Supplementary Note 4). Notably, in contrast to other SEB groups, both Pedi and Tsonga showed a strong *Ne* decline in the last 20 generations, which could be a function of cryptic relatedness. Therefore, we re-estimated the *Ne* profiles for these groups based only on unrelated participants with PIHAT < 0.05 (Fig. 3d). The filtering for relatedness removed the recent drop in population size observed in both populations. The *Ne* profile for Pedi participants after filtering also shows much higher resemblance to other Sotho-Tswana speakers. However, whether the related or the unrelated samples represent the actual demographic history of these SEB groups remains an open question for future studies.

We further partitioned the contribution of the two major source ancestries (Khoe-San and BS) to the *Ne* profiles of the SEB groups by using the ancestry-specific IBDNe approach[50]. The results depicted in Fig. 3e, f, and Supplementary Fig. 11 clearly show that the *Ne* curves, although being driven by BS ancestry, are also affected by Khoe-San gene flow. The Khoe-San ancestry impact on the *Ne* profiles was correlated with the level of Khoe-San ancestry in a group, for example higher in Tswana compared to Pedi (Fig. 3e, f). Moreover, the Khoe-San ancestry, when found to impact, seems to mainly affect *Ne* estimates older than 20 generations. However, the *Ne* estimates for older dates in this analysis depend on identification of short IBD segments. The reliable detection of these segments becomes challenging in cases where a particular ancestry has very low admixture proportions. Therefore, the predictions for Khoe-San ancestry could be less accurate compared to that for Bantu speakers for dates older than 50 generations.

**Impact of population structure on phenotype variation and association studies**. To explore the possible phenotypic

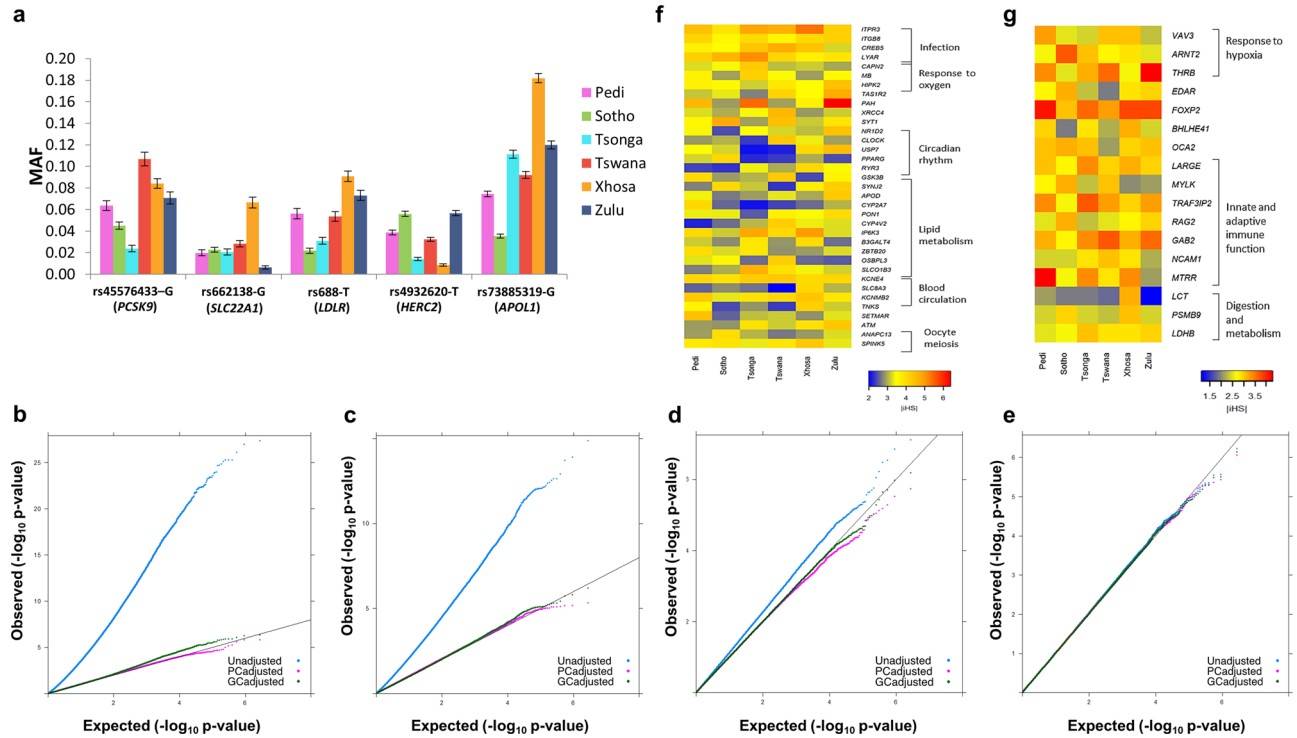

**Fig. 4 Possible impact of population structure within the South-Eastern Bantu-speaking (SEB) groups on genome-wide association studies (GWASs) and evolutionary estimates. a** Allele frequency variation of some of the well-known phenotype associated SNPs. The mean and the standard error was estimated using 50 random resampling runs with 30 samples each (source data provided in Source Data file). **b–e** Representative QQ plots showing results from simulated-trait GWASs comparing randomly sampled participants from **b** Agincourt (AGT) as cases to Soweto (SWT) as controls **c** 62.5% AGT + 37.5% SWT participants as cases to 100% SWT participants as controls. **d** Random samples from SWT without Tswana as cases to random samples from SWT with Tswana as controls. **e** Randomly sampled individuals from SWT as cases and controls. The Observed ($-log_{10}$ P-values) represent GWAS association results derived by logistic regression (two-tailed). The Expected ($-log_{10}$ P-values) are those based under the null hypothesis. For **b–e**, blue dots represent raw P-values, whereas purple and green dots represent P-values after principal component and genomic control based correction, respectively. **f** Heatmap showing differences in iHS statistics for some of the SNPs that were detected as outliers (|iHS| > 4; P-value < 0.003) in at least two of the SEB groups. **g** Heatmap showing differences in iHS statistics for SNPs in genes previously reported to be under positive selection, that were also detected to show moderate scores in one or more of the SEB groups (|iHS| > 3, P-value < 0.05).

implications of the fine-scale population structure, we compared allele frequencies of SNPs associated with various phenotypes (identified using the GWAS catalog) between the SEB groups. The comparison (Fig. 4a) shows almost four-fold variation in frequency of the *APOL1* variant rs73885319 among Sotho and Xhosa (MAF = 0.03 and 0.18, respectively). Similarly, alleles in genes such as *HERC2* (associated to skin colour), *PCSK9* and *SLC22A1* (associated to lipid level phenotype) also showed three-fold or higher allele frequency differences between SEB groups (Fig. 4a). A detailed list of 919 SNPs showing a minimum of three-fold difference in allele frequency is presented in Supplementary Data 1.

Population structure accompanied by high allele frequency differences could have major implications for genome-wide association studies (GWAS). Therefore, to assess the extent to which the observed structure could bias association results, we conducted four categories of simulated traits GWASs (binary trait) using study sites (AGT, DKG, and SWT) and/or ethno-linguistic labels as 'trait-proxies' (see Methods). Category 1 was aimed at stimulating a scenario where cases and controls are sampled from different study sites. The Fig. 4b shows a representative QQ plot for AGT-SWT (AGT as cases, SWT as controls) comparisons, which reflects a very strong population structure with exceptionally high (>4.5) genomic inflation scores. Category 2 represents a scenario where cases are randomly drawn from two sites (AGT and SWT), while controls were from one site only (SWT). The QQ plot for this category (Fig. 4c) shows that

even using about half of the samples from AGT could lead to substantially high (≥2) genomic inflation scores and large-scale deviations. Category 3 represents the situation when both cases and controls are drawn from the same site (SWT), but have preferential representation of SEB groups. Figure 4d shows that even ethno-linguistic stratification within a study site (SWT) resulted in the QQ curve reflecting population structure. Category 4 compares randomly assigned case and control status to individuals from the same site. As demonstrated in Fig. 4e, no major inflation was observed for this category.

The full results for 50 simulations (summarized in Supplementary Table 8) shows a substantial number of possible false positives associations in Categories 1 and 2, at the generally accepted genome-wide P-value threshold of $5 \times 10^{-8}$. While the genomic inflation normalizes with homogenization of the dataset, GWASs for category 3 and to a lesser extent category 4 generated false positives in a few simulations (Supplementary Table 8). Moreover, for each category, a substantial number of additional false-positive signals were detected at the suggestive P-value threshold of $5 \times 10^{-5}$, some of which, with slight changes in sample sizes could easily move below the genome-wide significance threshold (Supplementary Table 8). We also evaluated the extent to which the two standard approaches (genomic control based correction and PC-based correction using the first three PCs as covariates) can control genomic inflation and possible false positives in each category[52,53]. The results (Fig. 4b, e, and Supplementary Table 8) suggest that while both

approaches are effective, in some cases they fail to remove all the genome-wide significant associations due to population structure. Therefore, linear mixed model (with PC and kinship matrix as covariates) or other advanced approaches to address the population structure[52,54] could be more suitable for a GWAS involving SEB groups.

In order to reduce false positives due to small sample sizes, we restricted our simulations to only include common variants (MAF > 0.05). The addition of rare variants (MAF < 0.05) in a real GWAS, as well as increasing this dataset size by imputation, as is commonly performed in GWASs, could further increase false positives. Many of the signals from the simulated-trait GWASs have been previously reported as trait/disease genetic associations in the GWAS catalog[55] (Supplementary Data 2). Therefore, in a sample set containing an unbalanced (ethno-linguistically or geographically) proportion of SEB groups, the observed associations in a GWAS could give false associations resulting from intrinsic differences between these groups, rather than an association with the trait being investigated.

**Signatures of positive selection**. We used a haplotype homo-zygosity based selection scan to identify and compare outlier signals in the major SEB groups. The comparisons (Fig. 4f, g, Supplementary Data 3, Supplementary Fig. 12, and Supplementary Note 5) show several of these signals (in *SYT1, PAH, CAPN2,* and *SLC8A3* genes) to reach outlier threshold in some of the SEB groups but not in others, suggesting that the fine-scale structure can also influence evolutionary analyses. A population branch statistics (PBS) approach further detected SNPs showing high differentiation between Tswana and Tsonga (Supplementary Table 9 and Supplementary Note 5). Many of the SNPs showing outlier PBS scores mapped to immunity related genes such as *NFKBIE, VWF* and *ITGB2* (Supplementary Table 9).

**Preferential Khoe-San gene flow**. To study possible instances of preferential Khoe-San gene flow, we identified genomic regions deviating more than ±3 SD from the estimated average of Khoe-San ancestry in each SEB group. Despite the differences in the overall Khoe-San ancestry levels in these groups, we observed multiple genomic regions to show very high Khoe-San ancestry in more than one of the SEB groups (Supplementary Data 4 and Supplementary Fig. 13). For example, an extended region on chromosome 6 containing the *GRM4, HMGA1,* and *NUDT3* genes shows high Khoe-San ancestry in Pedi, Tsonga and Swazi, and part of this region was also observed to be Khoe-San enriched in Venda. Similarly, another region in chromosome 6 around the *TRDN* gene shows Khoe-San enrichment in Zulu and Xhosa. A Khoe-San ancestry region each in Tsonga (around *DAXX* and *ITPR3* genes) and Pedi (around *ZBTB20* gene) also harboured selection outliers hinting at possible post-admixture selection scenarios (Supplementary Data 4).

**Discussion**
More than 40 million South Africans speak one of the nine major South-Eastern Bantu languages as their first language. Notwithstanding clear divisions in the South-Eastern Bantu language phylogeny and geographic stratification of the speakers, very few studies have investigated the genetic differentiation between SEB groups. Based on a large-scale study of over 5000 participants representing eight of the nine major SEB groups in South Africa, we have demonstrated the presence of a robust fine-scale population structure within the SEB groups, which broadly separates genomes of SEB groups into the three major linguistic divisions (Nguni, Sotho-Tswana, and Tsonga), and also reflects the geographic distribution of LMAs to a large extent. The resolution of

this structure within the SEB groups was enhanced considerably by taking ethno-linguistic concordance of individuals and their geographic locations into account. However, it needs to be noted that self-identity itself is complex, with about one third of the participants having more than one parent or grand-parent with a different ethnic self-identity. Moreover, while the PCA and PCA-UMAP shows clear population structure, there are exceptions highlighting the fluidity of cultural identity. Thus, self-selected group-identity encompasses significant group-related genetic variability, and it is important to emphasise that cultural identity and genetic variation are not necessarily aligned. Studies on population structure in South Africa should not be seen as justifying the ethnic nationalism generated by the country's colonial and apartheid past. Our aim was to explore the role of genetic diversity in explaining population history and in health research. We recognise, and our study shows, that self-identity can involve considerable fluidity and that biological reductionist approaches pose dangers for the interpretation of our findings.

In alignment with results from previous studies[10,32], our data also shows that differential Khoe-San gene flow plays a major role in the population structure of SEB groups. However, the persistence of the structure even after accounting for differential Khoe-San admixture suggests the contribution of other demographic factors in the genetic differentiation of these groups. The SEB groups start to show clear divergence in population size dynamics from about 40 generations ago. This timeframe converges with the earliest dates of Khoe-San admixture and probably points at the initiation of migration events that gradually separated these groups. On the other hand, a rather wide variation in Khoe-San admixture dates (spanning ~20 generations) among SEB groups possibly reflects the complexity of the settlement of different parts of the country by the ancestral BS populations. Comparison of present-day SEB groups with Iron-Age farmer genomes provided evidence for genetic continuity in a geographic region in Central-East South Africa for at least the last 300–500 years. Our results, while attesting to the well-known pattern of Khoe-San female-biased gene flow, showed notable differences in the extent of this bias among different SEB groups demonstrating that the nature of interaction between Khoe-San and BS could have varied temporally and geographically.

The dataset we generated for this study has provided a much better contextualization for previously sequenced Iron-Age genomes from Southern Africa. The SEB are unique in Africa, as being among the very few populations that contain considerable gene flow from the Khoe-San. These data therefore are of major importance in terms of understanding the interaction between the Khoe-San and other Southern African populations. They will play an important role in providing insights through comparative analyses once more genetic data from hunter-gatherers and ancient genomes from this geographic region become available.

Our analyses including allele frequency comparisons, genome-wide scans for selection and Khoe-San ancestry distribution show the SEB groups to be highly diverged at certain genomic regions. Based on simulated-trait GWAS, we further illustrate that the fine-scale population structure within the SEB groups could impact a GWAS by introducing a large number of false positives. A combination of cautious study design to minimize geographic and ethno-linguistic biases and stringent measures for population structure correction is therefore recommended for GWASs involving SEB groups. Moreover, while GWAS can address the false positives introduced due to population structure using genomic control, PC or other approaches, it is impossible to identify and control for population structure in candidate gene studies. Therefore, utmost care should be taken during study design to ethnically and geographically homogenise samples in order to control for false positives in association studies using limited markers.

A major limitation of our study is that the sampling sites do not cover the full geographic spread of SEB groups in the country, possibly causing some of the groups to be suboptimally represented in our dataset. Nevertheless, our results suggest that we are at a critical point in history where the population structure is still observable with efficient sampling and in-depth ethno-linguistic characterization, even if it is gradually diminishing due to migration and intermingling between different SEB groups. We hope that our findings will motivate studies with larger sample sizes and wider geographic representation to help unravel the demographic events that contributed to the peopling of South Africa.

## Methods

**Sampling and genotyping procedures**. The volunteers included in this study were sampled across three study sites (Fig. 1a); Agincourt (AGT), Dikgale (DKG) and Soweto (SWT) under the Africa-Wits-INDEPTH partnership for genomic studies (AWI-Gen) project as part of the Human Heredity and Health in Africa (H3Africa) Consortium[56]. Of these SWT is urban, whereas DKG and AGT are rural/semi-urban sites. The study included a total of 5268 individuals (mostly within the age range of 40–60 years) representing eight major South African SEB groups: Tsonga, Pedi, Venda, Sotho, Tswana, Swazi, Zulu, and Xhosa. This study was approved by the Human Research Ethics Committee (Medical) of the University of the Witwatersrand (Wits) (protocol number M121029), and renewed in 2017 (protocol number M170880). In addition, research at the Dikgale Study Centre was approved by the Medunsa Research and Ethics Committee of the University of Limpopo (MREC/HS/195/2014:CR). Community engagement preceded sample collection and all participants provided broad consent for medical and population genetic studies. Details of community engagement, written informed consent, and genomic DNA extraction from blood samples have been described elsewhere[57]. In brief, the community engagement was tailored for each of the study sites according to their setting in a rural or urban area. It involved meetings with community leaders and elders, providing an opportunity to ask questions about the study and the potential benefits for the communities. The opportunity was also used to raise awareness of the rise in cardiometabolic diseases, as the AWI-Gen study aims to identify environmental and genetic risk factors. At the end of the first round of data collection there was community level and individual level feedback and further discussions. The outcomes of this population genetics study will be shared in future community interactions. The samples were genotyped on the H3Africa array (~2.3 M SNPs) using the Illumina FastTrack Sequencing Service2. The default Illumina pipeline was used for the genotype calling (build GRCh37/hg19).

**Language Majority area map**. The map in Fig. 1a represents the Language Majority Areas in South Africa. The map was redrawn from the original map, obtained from: https://en.wikipedia.org/wiki/Languages_of_South_Africa#/media/File:South_Africa_2011_dominant_language_map.svg. Attribution: "the user acknowledges Stats SA as the source of the basic data wherever they process, apply, utilise, publish or distribute the data, and also that they specify that the relevant application and analysis (where applicable) result from their own processing of the data". The language centroid points were calculated for this study by the author of the original map, Adrian Frith. The geometric medians of the population of speakers of each language (using Weiszfeld's Algorithm essentially as described at http://www.or.uni-bonn.de/~vygen/files/fl.pdf) were calculated. Since the large population sizes of the cosmopolitan Gauteng Province (containing many speakers of all languages) will distort the picture, geometric medians excluding Gauteng have been calculated.

**Data quality control procedures**. Quality control (QC) on the AWI-Gen geno-type dataset was performed using PLINK (v1.9)[58] and involved removal of duplicate SNPs, multi-allelic SNPs, INDELs and SNPs with a missingness >0.05, MAF < 0.01 and SNPs that failed HWE test (P-value < 0.0001). Individuals with missingness >0.05, discordant sex information and lacking self-reported ethnicity information were also removed. The genotype dataset post-QC consists of 5056 samples and 1,733,001 autosomal SNPs (AWI-S1) (Table 1 and Supplementary Table 1). A linkage disequilibrium (LD)-pruned version of this dataset was generated by removing SNPs in high LD ($r^2 > 0.5$ within a window of 50 SNPs, and with a window slide of 5 SNPs) using PLINK. The same parameters for LD-pruning were used for the datasets described below.

**Assessment of relatedness**. To identify related individuals, we estimated identity-by-descent (IBD) segments for each sample pair, based on the LD-pruned AWI-S1 dataset. For each pair of related individuals (PIHAT > 0.18), the sample with higher missingness was dropped, resulting in the removal of 737 SEB participants in the process leading to AWI-S2 dataset (Table 1). We also estimated genetic relatedness for all pairs of individuals in the AWI-S2 dataset using KING[59] and GENESIS[60] and PC-Relate option for plotting. After these QC-steps, no first-degree or second-degree relatives were found in the dataset used for the analyses below (Supplementary Fig. 14).

**Analysis of ethno-linguistic concordance**. In addition to self-reported ethnicity of the participant, the study also captured self-reported ethnicities of the parents and grandparents of each participant. Admixture within South-Eastern Bantu-speaking (SEB) groups as well as between SEB and non-SEB groups has been common in recent South African history. Since admixture events could influence fine-scale comparisons between SEB groups, we identified the participants that were ethno-linguistically concordant (EC), i.e., they have reported the same ethnicity for themselves, both parents and the four grandparents (allowing for a maximum of one mismatch). This set of 2,702 EC participants was defined as AWI-S3 dataset, details are listed in Table 1).

**Sample size homogenisation**. The representation of various SEB groups in the AWI-Gen study was notably skewed toward Tsonga, Pedi, and Zulu (with over 2000, 1200, and 600 samples, respectively) (Table 1). To avoid bias due to sample size differences and make the population sizes of the SEB groups comparable, we randomly downsized these three large groups to 80 individuals for each group from the AWI-S3 dataset. This dataset referred to as AWI-S4 consists of 476 samples (Supplementary Table 1).

**Data merging workflow**. For comparison of our population with previously studied populations, the AWI-S2 data (4,319 SEB unrelated samples) was merged with additional worldwide datasets from (ref. [21]), 1000 Genomes Project Phase 3 (KGP)[61], and African Genome Variation Project (AGVP)[32] using PLINK (hereafter Merged dataset 1), and only the SNPs that overlapped between all datasets were retained (Supplementary Table 1). We also generated another dataset (hereafter Merged dataset 2) by merging the above-mentioned dataset with data from Bantu-speaking groups in South Africa, e.g., the Southern African Human Genome Project (SAHGP)[31], and Mozambique[36] (Supplementary Table 1). In addition, the AWI-S3 dataset was also merged with five Iron-Age samples (AWI-AG dataset) with Bantu-related ancestry present in ancient DNA studies[10,11]. An additional dataset based on merging Khoe-San data[62] to AWI-S3 was generated for X-chromosome analysis (AWI-MV dataset). This layered merging was performed to retain the maximum number of SNPs possible for each analysis.

**Exploring population structure**. To investigate the population structure within the SEB groups, principal component analysis (PCA) was performed on the basis of the LD-pruned AWI-S2 dataset using the program smartPCA implemented in the EIGENSOFT suite[63]. Additionally, PCA was also performed first on the basis of the LD-pruned AWI-S3 dataset, and then for the Merged dataset 2. To further investigate the population structure obtained in PCA results, we combined the information for the first 10 PCs using a non-linear dimensionality reduction tool, called uniform manifold approximation and projection (UMAP)[34]. Lastly, to better discern fine-scale population structure among SEB groups we performed haplotype-based clustering analysis using ChromoPainter jointly with fineSTRUCTURE[35] (Supplementary Note 2).

**Genetic distance between SEB groups**. To investigate genetic affinities between the different SEB groups, we estimated Weir and Cockerham's $F_{ST}$ statistics (Weir and Cockerham 1984) between pairwise SEB populations included in the Merged dataset 2 (EC downsized) using PLINK. The relationship between the SEB groups based on pairwise $F_{ST}$ values was represented with a UPGMA tree using the program MEGA X[64].

**Linguistic phylogeny of SEB languages**. The linguistic phylogeny is based on lexical data for 100 concepts in 69 Bantu varieties, 34 of them part of South-Eastern Bantu languages and 35 outgroup languages belonging to different major Bantu branches[65]. The lexical data were binary recorded in 1304 partial cognate sets (form-meaning associations). The resulting matrix was analysed with Bayesian inference methods as implemented in MrBayes (v3.2.7)[66] using a restriction-site model[67].

**Correlations between geographic and genetic distance**. Procrustes transformation analysis and Mantel tests were implemented to investigate possible relationships between the geographic and genetic distances between the SEB groups. As many of the groups were sampled from sites that are quite distant to their native geography, such as Zulu and Xhosa, we calculated geometric medians of the population of speakers for each language using Weiszfeld's algorithm (http://www.or.uni-bonn.de/~vygen/files/fl.pdf), and considered them as the midpoints of each group. The great circle geographic distance between each midpoint was estimated using an online tool (https://www.geodatasource.com/distance-calculator). All individuals belonging to a particular SEB group were assigned to the same geographic location (geometric median), and the Procrustes transformation analysis was performed using the R package MCMCpack[68]. The correlations between PCA (PC1-PC2) results and geographic location of each SEB group was estimated using R package vegan[69] (9999 permutations). For the Mantel test, the genetic distance

matrix was based on weighted mean $F_{ST}$ estimates for each pair of SEB groups. This once again was performed using the R package *vegan*[69], using 9999 permutations to test the correlations between the geographic distances and $F_{ST}$-based genetic distances.

**Estimating admixture dynamics**. For global ancestry inference, we used an unsupervised clustering algorithm implemented in ADMIXTURE (v1.3)[37] on the Merged dataset 2 (EC downsized). The number of K-groups analysed varied from $K = 3$ to $K = 8$, and 50 independent runs with a random seed for each K-group was performed. The K-group with the lowest cross-validation error was considered "optimal". PONG[70] was used for merging and visualizing the clustering outputs of all the runs from the ADMIXTURE analysis, and major modes were used for the ADMIXTURE plots. To compare the differential contributions of the main ancestries (Khoe-San, BS and Eurasian component), the average admixture proportions of each ancestry was computed from the ADMIXTURE results at $K = 3$ for each SEB group in the Merged dataset 1. For each study site, we further estimated the average admixture proportion for each ancestry at $K = 3$ in each SEB group. We applied a *t*-test to compare whether there are significant differences in Khoe-San ancestry proportion across the three sites for a given SEB group.

**Local ancestry deconvolution**. For local ancestry inference, we used RFMix (v1.54)[71], on the basis of the Merged dataset 1 (EC). As reference panels, we selected: YRI for Central-West African ancestry; CEU for Eurasian ancestry; and combined Ju|'hoansi, /Gui //Gana, and Karretjie[21] for Khoe-San ancestry. The merged dataset was first phased using SHAPEIT2[72] with a reference panel of worldwide haplotypes[61], and then analysed with two runs of expectation maximization (EM = 2), forward-backward and PopPhased options. The genetic map from HapMap Phase 2 build GRCh37/hg19 was used for the analysis.

**Ancestry-specific PCA**. To investigate whether the differential Khoe-San gene flow is the only factor leading to the observed population structure, SEB haploid genomes were masked for regions of Khoe-San and European ancestries identified using RFMix. We then analysed haploid regions with more than 50% Bantu-related ancestry using the ancestry-specific PCA approach[73].

**Admixture date inference**. To reconstruct the timeframe of admixture events between the major ancestry components in SEB populations, we used three admixture dating methods, fastGLOBETROTTER, the recent implementation of GLOBETROTTER[38], MALDER (v1.0)[43], and MOSAIC (v1.3.7)[44]. The details for each method is described in Supplementary Note 2.

**Comparison with Iron-Age genomes**. To compare the genetic affinities of modern SEB groups to Iron-age Bantu-related samples from Southern Africa, we analysed the AWI-S3 dataset together with five ancient samples: four associated with Iron-Age (300–500 year old) farmers in South Africa[10], and one 700-years-old sample from Pemba, Tanzania (AWI-AG dataset)[11]. We used smartPCA to project the ancient samples onto the modern samples (using the following options: lsqproject = YES; killr2 = YES; and shrinkmode = YES). To better visualize genetic affinities between ancient and modern samples, we performed the PCA-UMAP analysis using the PC coordinates for the first ten PCs and UMAP tool for the analysis [34], and a custom Python script for the plotting.

**Y-chromosome and mitochondrial haplogroup analysis**. Y-haplogroup analysis was carried out using our new in-house *plotY* tool (https://github.com/shaze/ymthaplotools), based on a modified version of the tree and mutations table of AMY-tree[74]. The results were then validated using SNAPPY[75]. MtDNA haplotyping was performed using Haplogrep 2[76], using Phylotree mtDNA tree build 17rsrs-RSRS[77]. The details for mtDNA and Y-haplogroup detection are described in Supplementary Note 3.

**Sex-biased admixture patterns**. Recent literature has suggested the comparison of X-chromosome and autosomal contributions from the two source population as a robust method to test for possible sex-biased admixture[78]. To investigate the extent of sex-bias in the contributions of different ancestral populations to admixed SEB groups, the AWI-S3 and YRI and CEU from KGP were merged with available data[62], consisting of the 33 Khoe-San samples. The ancestry proportions for each were estimated using ADMIXTURE at $K = 3$ for three datasets: the X-chromosome dataset, the autosomal dataset, and the merged autosome-X-chromosome dataset. Admixture difference (ΔAdmix) ratios were then calculated using the method proposed by (ref.[79]). A positive ΔAdmix ratio indicates an excess of female-specific admixture contributions, while a negative value indicates an excess of male-specific admixture. To test statistical significance of the difference between the ΔAdmix for each ancestry between pairs of populations, we used the Wilcoxon rank-sum test.

**Population size dynamics**. To estimate and compare the variation in recent population size of the different SEB groups, IBD segments were detected from the downsized Merged dataset 1 using the program IBDseq[80] for each group (with

default parameters). The output was then used as input for the program IBDNe[50], which computes the effective population size ($Ne$) for each SEB group for the last few hundred generations. To avoid the conflation effect of short IBD segments[81], only IBD segments longer than 4 cM were retained for the $Ne$ estimation, and the remaining parameters were set as default. Ancestry-specific effective population size (ancestry-specific IBDNe)[50] was estimated for the different SEB groups to identify the contribution of Khoe-San and BS ancestries to the $Ne$ dynamics of various SEB groups. This analysis was performed on the dataset that was used to estimate the overall $Ne$. We followed the pipeline provided by the authors, which implements both IBD and local ancestry information from the genotype data. The first step in this approach was to phase the data using Beagle (v5)[82], and then IBD segments were detected using RefinedIBD and local ancestry information was inferred with RFMix (YRI, Khoe-San and CEU were used as reference source populations). Finally, IBDNe was used to estimate the ancestry-specific $Ne$ from the detected IBD segments and the ancestry blocks inferred from the local ancestry analysis.

**Allele frequency variation of phenotype associated variants**. We used PLINK to estimate allele frequencies of all SNPs in our dataset that are included in the GWAS catalog[55] (accessed on 19 April 2020), in the six major SEB groups (represented by at least 80 individuals in the AWI-S4 dataset). Standard error for allele frequencies was estimated using 50 bootstrap iterations in a subset of 30 individuals from each SEB group.

**Simulated genetic associations to illustrate the potential effect of population structure**. To simulate various possible scenarios for genetic association studies using ethno-linguistically and geographically mixed sets of SEB participants, four categories of artificial "case-control" trait simulations were performed. In the first category, the sampling site was used as the basis for assigning the case and control status. Here, the "case" label was assigned to 800 randomly sampled individuals from one of the three sites and the "control" label assigned to 800 randomly sampled individuals from a different site. Independent comparisons, 50 iterations each for AGT-DKG; DKG-SWT; AGT-SWT were performed. The second category corresponds to a scenario in which cases ($n = 800$) are a mixture of samples from AGT and SWT and the controls ($n = 800$) are sampled from SWT only. Three sets of cases with varying proportions of AGT and SWT representation (37.5% AGT + 62.5% SWT, 50% AGT + 50% SWT, and 62.5% AGT + 37.5% SWT) were generated and 50 iterations were performed for each set. The third category of trait simulation was aimed at studying the impact of ethno-linguistic stratification within a sampling site, SWT. For two sets (50 iterations, 500 cases–500 controls) generated in this category, the assignment was done in a way in which one of the ethno-linguistic groups (Tswana in set 1 and Tsonga in set 2) was absent in cases but present in controls. The fourth category was generated by randomly assigning case and control labels to the samples from a single site at a time. GWAS for each of the case-control pairs in all the sets under the four categories were conducted using the association testing function in PLINK. Genomic inflation scores were recorded for each run, and signals at a genome-wide significance threshold of $P$-value = $5 \times 10^{-8}$ were identified, as well as a less stringent suggestive significance threshold ($P$-value = $1 \times 10^{-5}$). To assess the extent of population structure correction possible with a genomic control based approach, for each run the inbuilt correction testing function was implemented using the "adjust" flag in PLINK. To assess the impact of PC-based correction, for each of the case-control iterations, PCs for the dataset were estimated using PLINK and the first three principal components were used as covariates in logistic regression based association testing in PLINK. QQ plots were generated using a custom R script. Possible phenotypic roles of the associations detected in these simulated-trait GWASs were assessed using the GWAS catalog[55].

**Genome-wide scans for selection**. To identify SNPs under positive selection, we calculated the integrated haplotype homozygosity scores (iHS)[83] implemented in the program *Selscan* (Szpiech and Hernandez, 2014). The AWI-S4 dataset was used for this analysis, and only SNPs with MAF < 0.05 were considered. We included six SEB groups, and we removed Venda and Swazi samples due to their small sample size. For each SEB group, the raw iHS were normalized across 40 frequency bins. A random sampling of scores across populations was performed to assess $P$-values for various score cutoffs. Based on this |iHS| > 4 was considered as outliers ($P$-value < 0.003). The mapping of SNPs to genes was performed based on information retrieved from Ensembl Biomart (Ensembl genes version 100; https://grch37.ensembl.org/biomart/). We also used the population branch statistics (PBS) analysis[84] to identify SNPs under positive selection. PBS is a summary statistic that utilizes pairwise Fst values among three populations to quantify genetic differentiation along each branch of their corresponding three-population tree. Since the overall genetic distance between the SEB groups is not very high, we considered only two groups from our study: the one with the highest Khoe-San ancestry (Tswana), and the other with the lowest Khoe-San ancestry (Tsonga). CHB from KGP was selected as the outlier population for this study. For each exonic SNP (identified using Ensembl Biomart as mentioned above) with MAF > 0.01, $F_{ST}$ values were estimated between the three pairs of the populations (CHB-Tswana, CHB-Tsonga, and Tswana-Tsonga) using VCFtools[85]. PBS scores were then

estimated in Tswana-Tsonga-CHB and Tsonga-Tswana-CHB comparisons using the method described in (ref. [84]).

**Preferential Khoe-San gene flow**. To identify genomic regions showing enrichment of Khoe-San ancestry in the SEB groups, local ancestry inference was performed using RFMIX as described above. To avoid statistical noise, regions around centromeres and telomeres (2 Mb from each side) for each chromosome were excluded from the analysis. Only SNPs with a high confidence value for Khoe-San ancestry (i.e., posterior probabilities value > 0.8) were retained for the analysis. Ancestry regions (containing at least 3 SNPs) exceeding the average genome-wide Khoe-San ancestry estimate by at least +3SDs were considered as candidates for preferential Khoe-San gene flow. We then investigated whether there are regions of adaptive introgression in the genomes by overlapping the regions under positive selection (as described above) and regions showing Khoe-San enrichment.

**Reporting summary**. Further information on research design is available in the Nature Research Reporting Summary linked to this article.

## Data availability

Genome-wide genotype data from the AWI-Gen study have been deposited in the European Genome-phenome Archive (EGA; https://ega-archive.org/) with the accession number: EGAD00010001996. DNA samples are archived in H3Africa biorepositories as part of the H3Africa Consortium agreement. Both data and biospecimens are available to interested researchers through EGA, subject to controlled access review by the Data and Biospecimen Access Committee of the H3Africa Consortium. Publicly available datasets included in the study are the following: 1000 Genomes Project Phase 3 (ftp:// ftp.1000genomes.ebi.ac.uk/vol1/ftp), SAHGP (https://www.ebi.ac.uk/ega/studies/ EGAS00001002639), Schlebusch et al, 2012 (http://jakobssonlab.iob.uu.se/data), AGVP (https://www.ebi.ac.uk/ega/studies/EGAS00001000960), Vicente et al., 2019 (https:// www.ebi.ac.uk/arrayexpress/experiments/E-MTAB-7813), Semo et al., 2019 (https:// www.ebi.ac.uk/arrayexpress/experiments/E-MTAB-8450), Schlebusch et al. 2017 (https:// www.ebi.ac.uk/ena/browser/view/PRJEB22660) and Skoglund et al 2017 (https://www. ebi.ac.uk/ena/browser/view/PRJEB21878). Source data for some of the figures and supplementary figures are provided with this paper (Source_data.xlsx). The source data for other display items are available on request.

## Code availability

All software and analysis code is publicly available. The code for plotY is available through GitHub (https://github.com/shaze/ymthaplotools). Codes for PCA-UMAP visualization and QQ plot generation are available on request.

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

## Acknowledgements

We wish to express our profound gratitude to the more than 5000 unnamed participants who took part in the study and provided blood samples, as well as the team of field workers, laboratory scientists and administrators who made the sample collection possible. We would like to acknowledge Adrian Frith for providing us with the coordinates of linguistic majority areas of each SEB group and Prof. Christopher Mathew for helpful comments and feedback about the manuscript. The AWI-Gen Collaborative Centre is funded by the National Human Genome Research Institute (NHGRI), Office of the Director (OD), Eunice Kennedy Shriver National Institute of Child Health & Human Development (NICHD), the National Institute of Environmental Health Sciences (NIEHS), the Office of AIDS research (OAR) and the National Institute of Diabetes and Digestive and Kidney Diseases (NIDDK), of the National Institutes of Health (NIH) under award number U54HG006938 and its supplements, as part of the H3Africa Consortium. D.S. and A.C. were supported by this grant. M.R. is a South African Research Chair in Genomics and Bioinformatics of African populations hosted by the University of the Witwatersrand, funded by the Department of Science and Technology and administered by National Research Foundation of South Africa (NRF). The Agincourt HDSS receives core support from the University of the Witwatersrand and the Medical Research Council, South Africa, and the Wellcome Trust, UK (Grant numbers 058893/Z/99/A; 069683/Z/02/Z; 085477/Z/08/Z; 085477/B/08/Z). The Birth to Twenty Cohort (Soweto, South Africa) is supported by the University of the Witwatersrand, the Medical Research Council, South Africa, and Wellcome Trust, UK. C.M.S. and C.F.-L. were funded by the European Research Council (ERC—no. 759933) as well as K.B. (ERC-CG—no. 724275). H.G. was supported by the Research Foundation Flanders (FWO, 12P8419N). This paper describes the views of the authors and does not necessarily represent the official views of the funders.

## Author contributions

Study design: D.S., A.C., C.M.S., M.R. and S.H., Data preparation and initial processing: S.H., Analysis: D.S., C.F.-L., A.C., S.A. and S.H., Archaeological, linguistic and historical data interpretation: G.W., H.G., N.C.-P., K.B. and P.D., Writing: D.S. and A.C. with contributions from all other authors, Other contributions: S.T., F.X.G.-O., S.N., F.M. and M.A. directed the field work and sample collection. All authors critically evaluated and approved the manuscript.

## Competing interests

The authors declare no competing interests.

## Additional information

## AWI-Gen Study

Dhriti Sengupta[1,15], Ananyo Choudhury[1,15], Shaun Aron[1], Stephen Tollman[8], F. Xavier Gómez-Olivé[8], Shane Norris[9], Felistas Mashinya[10], Marianne Alberts[10], Scott Hazelhurst[1,11], Michèle Ramsay[1,12,16 ✉]

## H3Africa Consortium

Dhriti Sengupta[1,15], Ananyo Choudhury[1,15], Shaun Aron[1], Stephen Tollman[8], F. Xavier Gómez-Olivé[8], Shane Norris[9], Felistas Mashinya[10], Marianne Alberts[10], Scott Hazelhurst[1,11], Michèle Ramsay[1,12,16 ✉]

