## [Peer Review File · Nature Communications]

Reviewer #1 (Remarks to the Author):

This manuscript does a good job of analyzing and presenting the presence of population substructure among southeastern Bantu-speaking populations of southern Africa. Additionally, it does the important job of demonstrating how this can affect biomedical research, while also increasing our sample size and knowledge about these underrepresented groups. Using the largest dataset to date for these groups, the authors show that population structure is largely correlated with the linguistic phylogeny and geography. They investigate this structure by analyzing differential Khoe-San gene flow, sex-biased gene flow, the timing of admixture, and changes in effective population size through time. Although Khoe-San ancestry varied greatly between the groups, the authors show that the observed structure is not merely the result of differential Khoe-San gene flow. Sex-biased gene flow, admixture timing, and changes in population growth also varied between the populations, emphasizing distinct demographic histories for the different groups. The authors also compare their dataset to five ancient (Iron Age) genomes which provide evidence for some genetic continuity in the region. Finally, these findings are considered in the context of future biomedical research. The authors simulate GWAS studies under four different sampling scenarios to investigate how the observed population structure could interfere with case-control sampling in GWAS studies and find that false positives can occur under multiple scenarios. We believe the authors do a good job of placing these findings in context and provide valuable information for future research.

A few minor notes:

1. IBDNe results: In Figure 3c, there are no confidence intervals in the Ne estimates for the different populations. The Brownings recommend cutting off IBDNe at 4g as the last few generations are unreliable, give the complexity of pedigree / demography inference from IBD. A $PI(\hat{a})$ of 0.18 is the expectation for second degree relatives, which are automatically removed in IBDNe. But I do like that they show the sensitivity to relatives in 3d. We are confused why the Pedi in 3d have an order of magnitude lower Ne than the Pedi in 3e (shouldn't this be the comparable black line?) It is suspicious that the SEB and KhoeSan ancestry have the same ancestral Ne at 100 g in these analyses. As they have very distinct population sizes prior to admixture (see many papers) there is no reason for them to converge. The only exception is the Tsonga which has the lowest KS ancestry - indicative of an artifact in the IBDNe program.
2. The manuscript mentioned the SNP density of the H3Africa array used for the AWI-Gen study; however, some of the analyses involved merging datasets from other studies and I did not see any mention of the SNP density for merged sets. This would be nice to know.
3. There are also a couple small typos. On page 9 in the first paragraph of the discussion there appears to be an inserted "to" in the phrase beginning: "which largely separates the genomes of SEB groups into...". On page 17 in the paragraph about the simulations when talking about the fourth category, it says "form" instead of "from."
4. We recommend replacing 'K-S' with Khoe-San throughout. There is no reason to abbreviate such a short name and it makes the manuscript approachable for non-specialists.
5. In Supp Figure 1 the population legends are switched between A and B.
6. As shown in Support Figure 4, the lowest CV error occurs at $k=4,5$. The main text figure should show all 3 k values. I suspect the 'European' ancestry in the YRI is due to the very unequal sample sizes of SEB compared to a single West African population. Authors should iterate with samples of approximately equal sizes for each major ancestry, as they do in the PCA. This may attenuate the Eurasian ancestry in the SEB as well.
7. Admixture dates: It is interesting to note that the time of Khoe-San gene flow into SEB here is relatively early compared to the reverse. Pickrell 2012 and Uren 2016, using 2 different methods, both found that Bantu → Khoe-San in the southern Kalahari occurred 10-15 generations ago. This should be mentioned in the text. This may suggest that the Tswana first moved along the eastern edge of the Kalahari incorporating local Khoe-San groups that no longer persist, and only later moved into the arid regions of Kalahari encountering Ui-Taa speakers (at least ~300 y later).
8. We may have missed it, but UMAP results are not discussed in the main text. They should be moved to the supplement.
9. There is a whole paper on selection on DARC in Africa (w/ detailed discussion of the Zulu and San) - McManus PLoS Genetics 2017. Could the authors include this in their iHS results and discussion?

Reviewer #2 (Remarks to the Author):

The manuscript "Genetic-substructure and complex demographic history of South African Bantu speakers" but Sengupta et al. deals with the genome-wide analysis of more than 5,000 Bantu-speaking individuals in South Africa in order to unravel population structure and its consequences.

The manuscript is nicely written, clear in the presentation of the data and results, structured in the format, and with a clear goal. In addition, the present study provides a large data set that will be an invaluable resource for future genetic studies.

I have some comments about the manuscript that the author might consider addressing.

In the introduction section, a linguistic classification of South Eastern Bantu (SEB) languages is presented, but it would also be helpful for the readers to provide references for this linguistic classification of languages and indicate if there is consensus on this classification.

One of the main results of the manuscript is the finding of genetic substructure of the SEB groups. This is shown in the visual approach of the PCAs but it would be nicer to provide other evidence of the genetic structure. Perhaps the authors should consider refined methods such as the amount of haplotype sharing and haplotype clustering within and among SEB groups. This will provide further support to the genetic structure detected in the PCAs.

The correlation found between genetics and geography is tested through a Mantel test of Fst distances and geography. I am wondering if the authors could provide also a correlation between PCAs and geography since they base their main structure conclusions in the PCA (the correlation between Fst and PCAs is usually high so I do not expect any departures in the correlation).

When presenting the Admixture results in the text, it is stated African in blue, K-S in green, and European in red. I would suggest removing the "African" label for "Bantu" or other term that should not be misleading.

In Supplementary Figure 1, legend symbols in a and b should be checked

Reviewer #3 (Remarks to the Author):

This is an interesting article looks at the population structure of the South Eastern Bantu-speaking (SEB) groups that align with geography and linguistics of these groups. They also highlight the importance of accounting for population structure in biomedical genomic research.

I think this article is timely considering the discussions and focus on mixed and non-European ancestry biomedical genetic research.

I just have a few minor suggestions and questions.

1. Lines 146-147: "We reject biological reductionist interpretations of this work". This comes off a bit strong and also seems broad? Also, I'm not really sure what sort of interpretations you think might come from it?
2. Supplementary Figure 1: Did the group legend for a. and b. get switched? Looks like Ju|'hoansi, /Gui //Ghana and Karretjie are on a. not b.
3. Lin 335: These 919 SNPS look like they are mainly SNPs based on results in Europeans. What about findings from samples of African ancestry? How do they look across the different SEB groups?
4. How do the effect estimates of these 919 SNPs and any from African ancestry findings look in the SEB groups? Do they generalize well in some groups but not others?
5. Lines: 363-369: How many PCs did you adjust for? I looked in the methods too but maybe I just missed it?
6. What do you think about adding in the 1000 genomes reference panel populations, particularly the different AFR groups, to a set of the PCA plots? I would be interested to see how they look along with the SEB group.

REVIEWER COMMENTS

Reviewer #1 (Remarks to the Author):

This manuscript does a good job of analyzing and presenting the presence of population substructure among southeastern Bantu-speaking populations of southern Africa. Additionally, it does the important job of demonstrating how this can affect biomedical research, while also increasing our sample size and knowledge about these underrepresented groups. Using the largest dataset to date for these groups, the authors show that population structure is largely correlated with the linguistic phylogeny and geography. They investigate this structure by analyzing differential Khoe-San gene flow, sex-biased gene flow, the timing of admixture, and changes in effective population size through time. Although Khoe-San ancestry varied greatly between the groups, the authors show that the observed structure is not merely the result of differential Khoe-San gene flow. Sex-biased gene flow, admixture timing, and changes in population growth also varied between the populations, emphasizing distinct demographic histories for the different groups. The authors also compare their dataset to five ancient (Iron Age) genomes which provide evidence for some genetic continuity in the region. Finally, these findings are considered in the context of future biomedical research. The authors simulate GWAS studies under four different sampling scenarios to investigate how the observed population structure could interfere with case-control sampling in GWAS studies and find that false positives can occur under multiple scenarios. We believe the authors do a good job of placing these findings in context and provide valuable information for future research.

We thank the reviewer for their positive comments about our work. Throughout the manuscript we have used blue fonts to indicate the insertions and red strikethroughs to indicate the removed words.

A few minor notes:

1. **IBDNe results.** In Figure 3c, there are no confidence intervals in the N_e estimates for the different populations.

Response: We thank the reviewer for this observation, we have updated the figures (Figures 3c, d) following the reviewer's suggestion.

2. The Brownings recommend cutting off IBDNe at 4g as the last few generations are unreliable, given the complexity of pedigree / demography inference from IBD.

Response: We agree with the reviewer. We have also amended this on the figures (Figures 3c, f).

3. A $PI(\hat{a})$ of 0.18 is the expectation for second degree relatives, which are automatically removed in IBDNe. But I do like that they show the sensitivity to relatives in 3d. We are confused why the Pedi in 3d have an order of magnitude lower N_e than the Pedi in 3e (shouldn't this be the comparable black line?)

Response: We sincerely thank the reviewer for this careful observation. The difference in the plot was due to the difference in the scale used for the y-axis. We have amended the plots (Figures 3d, e) in the panels, so that they are now on the same scale.

4. It is suspicious that the SEB and KhoeSan ancestry have the same ancestral N_e at 100 g in these analyses. As they have very distinct population sizes prior to admixture (see many papers) there is no reason for them to converge. The only exception is the Tsonga which has the lowest KS ancestry - indicative of an artifact in the IBDNe program.

Response: We agree with the reviewer and thank them for flagging this important issue. Short IBD segments which represent coalescent events that occurred further back in time (50-100 generations ago), are more difficult to estimate due to false-positives. Moreover, in our IBDNe analysis we used a cut-off of 4cM to remove short IBD segments. Thus, N_e estimates of over 50 generations for Khoe-

San ancestry, could indeed be less accurate. We have therefore added the following line in the main text to clarify this:

“However, the N_e -estimates for older dates in these analyses depend on identification of short segments. The reliable detection of these segments become challenging in cases where a particular ancestry is not very high in proportion. Therefore, the predictions for Khoe-San ancestry could be less accurate compared to that for Bantu-speakers for dates older than 50 generations”.

5. The manuscript mentioned the SNP density of the H3Africa array used for the AWI-Gen study; however, some of the analyses involved merging datasets from other studies and I did not see any mention of the SNP density for merged sets. This would be nice to know.

Response: The information about SNP density and merging workflow was already provided in Supplementary Table 13, cited in the beginning of Methods.

6. There are also a couple small typos. On page 9 in the first paragraph of the discussion there appears to be an inserted “to” in the phrase beginning: “which largely separates the genomes of SEB groups into...”. On page 17 in the paragraph about the simulations when talking about the fourth category, it says “form” instead of “from.”

Response: Thanks for pointing this out. We have amended these.

7. We recommend replacing ‘K-S’ with Khoe-San throughout. There is no reason to abbreviate such a short name and it makes the manuscript approachable for non-specialists.

Response: In agreement with the reviewer’s suggestions we have changed all the K-S to Khoe-San in the main text.

8. In Supp Figure 1 the population legends are switched between A and B.

Response: We thank the reviewers for noticing this. We have amended the figure accordingly.

9. As shown in Support Figure 4, the lowest CV error occurs at $k=4,5$. The main text figure should show all 3 k values. I suspect the ‘European’ ancestry in the YRI is due to the very unequal sample sizes of SEB compared to a single West African population. Authors should iterate with samples of approximately equal sizes for each major ancestry, as they do in the PCA. This may attenuate the Eurasian ancestry in the SEB as well.

Response: We specially thank the reviewer for their careful scrutiny of the figure and their valuable inputs. Based on the reviewer’s suggestion we have added the plot for $K=4$ to the main figure (**Figure 2a**). Also, as advised, we have added a new ADMIXTURE analysis (**Supplementary Figure 4b**), that is based only on some selected SEB populations in order to homogenise the sample sizes across East, West and South Africa. This modification indeed removes the Eurasian ancestry in the YRI. We also added text referring to the figure in the main manuscript:

“To investigate the impact of differential sample sizes from Eastern, Western and Southern Africa in these estimates, we also repeated the ADMIXTURE analysis using a dataset with a more uniform representation of samples from the three regions (**Supplementary Fig. 4b**).”

10. Admixture dates: It is interesting to note that the time of Khoe-San gene flow into SEB here is relatively early compared to the reverse. Pickrell 2012 and Uren 2016, using 2 different methods, both found that Bantu → Khoe-San in the southern Kalahari occurred 10-15 generations ago. This should be mentioned in the text. This may suggest that the Tswana first moved along the eastern edge of the Kalahari incorporating local Khoe-San groups that no longer persist, and only later moved into the arid regions of Kalahari encountering Ui-Taa speakers (at least ~300 y later).

Response: We thank the reviewer for drawing our attention to this. We did not cover these studies in our discussion because of our focus on Eastern South Africa. However, we completely agree that comparison to these studies is important, and therefore added the following line to the relevant section

“The suggested dates for BS gene-flow into Khoe-San populations, especially in the southern Kalahari region appears to be much younger (10-15 generations ago)^{45,46} compared to the estimated dates for Khoe-San gene flow into the SEB. These large scale differences in dates indicate the possibility of independent migration and admixture dynamics of BS in eastern and western South Africa.”

11. We may have missed it, but UMAP results are not discussed in the main text. They should be moved to the supplement.

Response: We would like to draw the reviewers’ attention to line 165 and 267 (please see below) as well as to the UMAP plots in both Figure 1 and Figure 2 panels.

In Line 165: “PCA-UMAP analysis³⁰, based on a composite of the first 10 PC coordinates estimated using EC participants, further illustrates the separation between the SEB groups (**Fig. 1d**).”

In Line 267: “The PCA and PCA-UMAP projecting five Iron Age Bantu-related genomes (300 to 700 years old) onto the genetic variation.”

12. There is a whole paper on selection on DARC in Africa (w/ detailed discussion of the Zulu and San) - McManus PLoS Genetics 2017. Could the authors include this in their iHS results and discussion?

Response: We thank the reviewers for suggesting to this interesting publication. To keep the paper within word limits we have attempted to restrict the section describing signatures of selection to a minimum possible length. For the same reason, we discussed the importance of the DARC region in our results in the Supplementary Note 5-“Signatures of positive selection in SEB groups” and cited the McManus et. al. paper.

Reviewer #2 (Remarks to the Author):

The manuscript “Genetic-substructure and complex demographic history of South African Bantu speakers” but Sengupta et al. deals with the genome-wide analysis of more than 5,000 Bantu-speaking individuals in South Africa in order to unravel population structure and its consequences. The manuscript is nicely written, clear in the presentation of the data and results, structured in the format, and with a clear goal. In addition, the present study provides a large data set that will be an invaluable resource for future genetic studies.

We thank the reviewer for their positive comments about our work. Throughout the manuscript we have used blue fonts to indicate the insertions and red strikethroughs to indicate the removed words.

I have some comments about the manuscript that the author might consider addressing.

1. In the introduction section, a linguistic classification of South Eastern Bantu (SEB) languages is presented, but it would also be helpful for the readers to provide references for this linguistic classification of languages and indicate if there is consensus on this classification.

Response: We thank the reviewer for drawing our attention to this. In supplementary Note 1 we already delve into the “Linguistic phylogeny of the South Eastern Bantu languages”. However, this supplementary note was not referred to in the Introduction. Based on the reviewer’s suggestion we have modified the line, added four citations and also referred to the Supplementary note as :

“South Africa has 11 official languages of which nine are Bantu languages belonging to this family’s South Eastern branch. Within these nine languages two large sub-clusters are traditionally distinguished: Nguni (including Zulu, Xhosa, Swazi, and Ndebele) and Sotho-Tswana (including Sotho, Tswana, and Pedi), while Venda and Tsonga tend to be seen as independent linguistic entities

(Jones-Phillipson et al. 1972; Wentzel 1981, Doke 1954; Herbert et al. 2020). An entirely new lexicon-based linguistic phylogeny included in this study broadly confirms the traditionally recognized clusters, but also adds new insights into how these languages relate to each other as well as to 60 other Bantu languages from southern Africa and beyond (**Supplementary Note 1**.)”

References Added:

Doke, C.M. 1954. *The Southern Bantu Languages*. London: Oxford University Press for the International African Institute.

Herbert, Robert K. & Richard Bailey. 2002. *The Bantu languages: Sociohistorical perspectives*. In Mesthrie, Raj (ed.), *Language in South Africa*, 50-78. Cambridge: Cambridge University Press.

Jones-Phillipson, Rosalie. 1972. *Affinities between Venda and other Southern Bantu languages*. London: University of London PhD Thesis.

Wentzel, Petrus Johannes. 1981. *The relationship between Venda and Western Shona*. Pretoria: University of South Africa PhD Thesis.

We have also amended the Supplementary Note 1 to increase the clarity.

2. One of the main results of the manuscript is the finding of genetic substructure of the SEB groups. This is shown in the visual approach of the PCAs but it would be nicer to provide other evidence of the genetic structure. Perhaps the authors should consider refined methods such as the amount of haplotype sharing and haplotype clustering within and among SEB groups. This will provide further support to the genetic structure detected in the PCAs.

Response: We have now performed haplotype-based analysis using ChromoPainter/fineSTRUCTURE to estimate the haplotype sharing patterns among SEB individuals included in the AWI-S4 dataset (see new **Supplementary Figures 2c** and **2d**). Both, haplotype-based PCA and pairwise-coincidence matrix are in strong agreement with our previous results, highlighting population structure among SEB groups. Additional comments about the results and analysis were included in the Results and Methods sections, and additional details in the updated Supplementary Note 2.

In Results - “Fine-scale population structure within SEB” section:

“To avoid the likely influence of sample size-bias, we randomly downsized each group (AWI-S4 dataset). Likewise, both the PC and PCA-UMAP plots for this downsized data largely retained the fine-scale structure within SEB groups (**Supplementary Fig. 2a, b**). In addition, we performed haplotype-based analysis on the basis of the AWI-S4 dataset using ChromoPainter/fineSTRUCTURE³⁵ (see **Supplementary Note 2**). Haplotype-based PCA and the pairwise-coincidence matrix among EC individuals provides further support for the fine-scale population structure among SEB groups (**Supplementary Fig. 2c, d**).”

In Methods - “Exploring population structure” section:

“Lastly, to better discern fine-scale population structure among SEB groups we performed haplotype-based clustering analysis using ChromoPainter jointly with fineSTRUCTURE³⁵ (**Supplementary Note 2**).”

3. The correlation found between genetics and geography is tested through a Mantel test of Fst distances and geography. I am wondering if the authors could provide also a correlation between PCAs and geography since they base their main structure conclusions in the PCA (the correlation between Fst and PCAs is usually high so I do not expect any departures in the correlation).

Response: We thank the reviewer for this suggestion and have added results from a Procrustes transformation analysis as **Supplementary Figure 3c**. The results, legend of supplementary Figure 3 and the method sections have also been updated accordingly.

Modification in Results (Last paragraph of the section “Fine-scale population structure within SEB”):

“Phylogenetic trees based on genetic distances (FST) (**Fig. 1e and Supplementary Fig. 3a**) and linguistic phylogeny (**Fig. 1f, Supplementary Fig. 3b and Supplementary Note 1**) of the SEB groups shows overall alignment in topology. Similarly, the genetic (FST) and geographical distances between the SEB groups also show a moderate correlation (Mantel test r value: 0.56, P -value=0.002). Procrustes transformation analysis further highlights the correlation between PC and geography ($r^2=0.72$; P -value=0.0009) (**Supplementary Fig. 3c**). However, the overall low magnitude of FST values (**Supplementary Fig. 3e–3d**) suggests that the fine-scale structure, although robust, corresponds to relatively small genetic distances.”

4. When presenting the Admixture results in the text, it is stated African in blue, K-S in green, and European in red. I would suggest removing the “African” label for “Bantu” or other term that should not be misleading.

We thank the reviewer for the comment. The label has been changed accordingly.

5. In Supplementary Figure 1, legend symbols in a and b should be checked

We thank the reviewer for pointing this out. The legend has been modified accordingly.

Reviewer #3 (Remarks to the Author):

This is an interesting article looks at the population structure of the South Eastern Bantu-speaking (SEB) groups that align with geography and linguistics of these groups. They also highlight the importance of accounting for population structure in biomedical genomic research. I think this article is timely considering the discussions and focus on mixed and non-European ancestry biomedical genetic research. I just have a few minor suggestions and questions.

We thank the reviewer for their positive comments about our work. Throughout the manuscript we have used blue fonts to indicate the insertions and red strikethroughs to indicate the removed words.

1. Lines 146-147: “We reject biological reductionist interpretations of this work”. This comes off a bit strong and also seems broad? Also, I’m not really sure what sort of interpretations you think might come from it?

Response: Thank you for pointing out that this statement may be confusing to some readers and needs more context. Of course South Africa’s history of racism has made us very wary of how the outcomes of this study may be misinterpreted and misused. We have changed it as follows:

“Studies on population structure in South Africa should not be seen as justifying the ethnic nationalism generated by the country’s colonial and apartheid past. Our aim is to explore the role of genetic diversity in explaining population history and in health research. We recognise, and our study shows, that self-identity can involve considerable fluidity and that biological reductionist approaches pose dangers for the interpretation of our findings.”

2. Supplementary Figure 1: Did the group legend for a. and b. get switched? Looks like Ju|’hoansi, /Gui //Ghana and Karretjie are on a. not b.

Response: We thank the reviewer for pointing this out. The figure has been updated accordingly.

3. Lin 335: These 919 SNPS look like they are mainly SNPs based on results in Europeans. What about findings from samples of African ancestry? How do they look across the different SEB groups?

Response: We welcome the suggestion and the variants detected in African GWAS have now been highlighted in Supplementary Table 7 and the table legend has been edited accordingly.

4. How do the effect estimates of these 919 SNPs and any from African ancestry findings look in the SEB groups? Do they generalize well in some groups but not others?

Response: We thank the reviewer for this interesting proposition. However, as we only used simulated traits (basically site of collection and ethnicity) instead of real phenotypes, unfortunately, we do not have the option of investigating the variation in effect sizes meaningfully.

5. Lines: 363-369: How many PCs did you adjust for? I looked in the methods too but maybe I just missed it?

Response: Although this is mentioned in the methods (line 826)- “PCs for the dataset were estimated using PLINK and the first three principal components were used as covariates in logistic regression based association testing in PLINK.”, we agree that this needs to be included in the main text as well. Therefore, we have added the line in the main text as follows :

“We also evaluated the extent to which the two standard approaches (genomic control (GC)-based correction and PC-based correction using the first three PCs as covariates, can control genomic inflation and possible false positives in each category.”

6. What do you think about adding in the 1000 genomes reference panel populations, particularly the different AFR groups, to a set of the PCA plots? I would be interested to see how they look along with the SEB group.

Response: We have added **Supplementary Figure 1c** that compares South African populations to other African populations using a PC. We have also added a line to the legend -

“(c) As expected, on PC1 South African populations from our study split from Eastern and Western African populations from the 1000 Genomes Project and the AGVP, while on PC2 the SEB samples with the highest Khoe-San admixture split from other SEB and African groups”.

Reviewer #1 (Remarks to the Author):

The authors have adequately addressed all comments.